# Volatile and Non-Volatile Characterization of White and Rosé Wines from Different Spanish Protected Designations of Origin

**Rubén Del Barrio-Galán** * , **Héctor del Valle-Herrero, Marta Bueno-Herrera, Pedro López-de-la-Cuesta and Silvia Pérez-Magariño** *

Instituto Tecnológico Agrario de Castilla y León, Consejería de Agricultura y Ganadería, Ctra Burgos Km 119, Finca Zamadueñas, 47071 Valladolid, Spain; valherhe@itacyl.es (H.d.V.-H.); bueherma@itacyl.es (M.B.-H.); lopcuepe@itacyl.es (P.L.-d.-l.-C.)

* Correspondence: rdelbarriogalan@gmail.com (R.D.B.-G.); permagsi@itacyl.es (S.P.-M.); Tel.: +34-983-415245 (S.P.-M.)

**Abstract:** The quality of wines has often been associated with their geographical area of production, as well as the grape variety used in their elaboration. Many research studies have been carried out to characterize and differentiate between red wines labeled with Protected Designation of Origin (PDO) from different geographical areas, but very few have been carried out on white and rosé wines. The objective of this work was to characterize white and rosé PDO wines from different geographical areas of Spain very close to each other elaborated with different grape varieties and select the variables that most contribute to their differentiation. Analysis of variance (ANOVA) and principal component analysis (PCA) were used as statistical methods. The ethanol content was the nonvolatile variable that most contributed to differentiating between some of the white and rosé wines according to their PDO. The white wines from RD (Ribera del Duero) and BI (Bierzo) were characterized by a high terpenic content (floral notes) while the wines from RU (Rueda), TO (Toro) and CI (Cigales)by a high content of ethyl esters and alcohol acetates (fruity aromas). The rosé wines elaborated with the Mencía grape variety from BI were characterized by their highest polysaccharidic content, which could have a positive sensory effect on the mouthfeel. The rosé wines from CI were characterized by their volatile profile complexity, having the highest content of volatile compounds from the oak wood, terpenes and C6 alcohols which provide pleasant woody, floral and herbaceous aromas. On the contrary, the RD wines were richest in alcohol acetates responsible for fruity aromas.

**Keywords:** wine differentiation; phenols; volatile compounds; polysaccharides; geographical origin

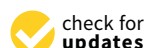

## 1. Introduction

Phenolic and volatile compounds have been the most common variables studied in wines due to their importance in chemical, physical and sensory properties [1–4]. Phenolic compounds come from grapes and oak wood of the barrels where wines are aged and affect color and gustatory properties of wines, such as astringency, bitterness and structure [1,2]. On the other hand, volatile compounds affect the olfactory quality of wines, which can come from grapes and the fermentation and aging processes of wines which affect the fruity, floral, herbaceous and toasted notes [3,4]. Other molecules, such as polysaccharides, have also been studied in the recent years because they have gained interest, mainly due to their influence in the olfactory and gustatory sensory phases of wines [5–7]. These compounds are usually grouped according to their origin, mainly, grapes and yeasts [8] and, to a lesser extent, those that come from oak wood barrels used for the aging process [9]. Organic acids, glycerol and ethanol content as well as pH and total acidity can influence the gustatory sensory properties of wines such as acidity, sweetness, body, bitterness and astringency [10–13].

All these compounds can vary largely in wines due to several factors such as environmental characteristics of the geographical region, grape varieties used in the winemaking,

vineyard location, the fermentation yeast strain used as well as local know-how applied in the winemaking [14–17]. Thus, wines labeled with Protected Designation of Origin (PDO) are characterized by particular physicochemical and sensory properties which may allow differentiating these wines from those elaborated in other geographical areas. Nowadays, consumers consider the origin of wines to be one of the most important factors when buying a wine, as well as other factors such as price, grape variety and wine category that might earn the consumer's liking for a wine [18,19].

Several studies carried out in different regions of several countries have shown that the composition of wines (volatile and nonvolatile compounds) can be very different depending on the aspects mentioned above [14,15,20–24]. Spain is one of the main wine-producing countries in the world with seventy-five recognized PDOs. The Castile and León region located in the North of Spain is one of the most important winemaking regions with thirteen PDOs, many of them very close geographically. However, it is not easy to differentiate between their wines due to the proximity between PDOs and also because the same grape variety is used to elaborate the wines in many of them. Therefore, the objective of this work was to characterize white and rosé wines from the most important PDOs of Castile and León (Ribera del Duero, Rueda, Toro, Bierzo and Cigales) and select the variables that most contribute to their differentiation.

## 2. Material and Methods

### 2.1. Wine Samples

Seventy-three commercial white and rosé wines provided by the different regulatory councils were analyzed. Table 1 shows the number of used wines from each PDO and grape variety since each regulatory council authorizes the use of certain varieties. White and rosé wines produced within these PDOs must be exclusively elaborated with the grape varieties established as principals and with a minimum percentage of each principal variety.

**Table 1.** Number of wines analyzed in the different PDOs and the minimum percentage of each variety (in parenthesis).

| GRAPE VARIETY | PDO | | | | | |
|---|---|---|---|---|---|---|
| **White wines** | **Rueda** | **Cigales** | **Toro** | **Ribera del Duero** | **Bierzo** | **Total** |
| **Verdejo** | 12 (85%) | 6 (50%) | 5 (85%) | | | 23 |
| **Sauvignon blanc** | 9 (85%) | | | | | 9 |
| **Godello** | | | | | 9 (85%) | 9 |
| **Malvasía castellana** | | | 4 (85%) | | | 4 |
| **Albillo mayor** | | | | 4 (75%) | | 4 |
| **Total** | 21 | 6 | 9 | 4 | 9 | 49 |
| **Rosé wines** | **Rueda** | **Cigales** | **Toro** | **Ribera del Duero** | **Bierzo** | **Total** |
| **Tempranillo** | | 10 (50%) | 4 (50%) | 4 (50%) | | **18** |
| **Mencía** | | | | | 6 (50%) | **6** |
| **Total** | | **10** | **4** | **4** | **6** | **24** |

### 2.2. Reagents and Standards

Phenolic compound standards were supplied by Sigma-Aldrich (Steinheim, Germany); Fluka (Buchs, Switzerland) and Extrasynthèse (Lyon, France). Polysaccharide standards were provided by Sigma-Aldrich (Steinheim, Germany).

Volatile compound standards were purchased from Fluka (Buchs, Switzerland), Sigma-Aldrich (Steinheim, Germany) and Lancaster (Strasbourg, France).

Ethanol (96%) was from Labkem (Barcelona, Spain). Acetonitrile and methanol for HPLC analyses were supplied by Carlo Erba (Sabadell, Spain), the remaining reagents—by Panreac (Madrid, Spain). Milli-Q water was obtained through a Millipore system (Bedford, MA, USA).

### 2.3. Analytical Methods

Classical enological parameters were determined according to the official methods of OIV [25].

Total organic acids (tartaric, malic and lactic acid) and glycerol (both expressed in g L$^{-1}$) were analyzed according to the methodology described in [26], with some modifications [15], and using a high-performance liquid chromatography (HPLC) column coupled to a diode array detector (DAD) and a refractive index detector (RID).

Color intensity was analyzed according to Glories [27]. Total polyphenols (TP) (expressed in mg L$^{-1}$ of gallic acid) were analyzed following the methodology described in [28]. Total tannins (TT) were analyzed using the methodology described in [29] based on tannin precipitation with methylcellulose and expressed in mg L$^{-1}$ of catechin. These parameters were measured using a UV/Vis Agilent Cary 60 spectrophotometer (Santa Clara, CA, USA).

Low-molecular-weight phenolic compounds (LMWPC) were analyzed by direct injection using an HPLC column coupled to a DAD system following the chromatographic conditions described in [30]. The individual phenolic compounds were grouped as follows: hydroxybenzoic acids (HBA), hydroxycinnamic acids (HCA), hydroxycinnamic acid tartaric esters (HCATE), flavanols, flavonols and phenolic alcohols. These compounds were expressed in mg L$^{-1}$ of the corresponding phenolic standard.

Higher alcohols were analyzed following the method described in [31], using a gas chromatograph with a flame ionization detector (GC-FID), and were expressed in mg/L of the corresponding standard.

Volatile compounds were extracted by headspace solid-phase microextraction (HS-SPME) following the methodology described in [32], with some modifications. For each sample, 8.5 mL wine and 50 µL internal standard solution (ISS, a mixture of 17.7 mg L$^{-1}$ methyl-2-methylbutyrate, 20 mg L$^{-1}$ benzyl alcohol $_{13}$C6, 45 mg L$^{-1}$ methyl octanoate, 185 mg L$^{-1}$ heptanoic acid, 20 mg L$^{-1}$ 3,4-dimethylphenol and 16.3 mg L$^{-1}$ hexanal; the internal standards used were not present in the studied samples) were diluted to a final volume of 25 mL with a hydroalcoholic solution (13.5% ethanol + 3.5 g L$^{-1}$ tartaric acid and pH adjusted to 3.5). After that, 10 mL of this dilution were put into a 20-mL glass vial with 3.5 g sodium chloride. The SPME fiber was inserted in the headspace of the sample vial and maintained for 60 min at 40 °C in agitation at 500 rpm for the extraction process. After that, the fiber was desorbed in the injector for 3 min at 240 °C in the splitless mode. Chromatographic analyses were performed with a gas chromatograph coupled to a quadrupole mass detector, equipped with a DB-WAX Ultra Inert capillary column (60 m × 0.25 mm × 0.50 µm), and following the chromatographic conditions established in [31]. The electron ionization mass spectra (40–200 amu) were acquired in the SIM/SCAN mode at 70 eV. Identification of the volatile compounds was carried out using the mass spectra of the calibration standards, retention times, and the NIST library. Quantification was carried out with calibration curves using the chemical standards of each of the compounds to be determined in the concentration range of application of the method and adding a known concentration of six internal standards (IS). Quantification of each compound was carried out in the SIM mode with the area of the quantification ion fixed with respect to one of the internal standards. The quantified compounds were grouped as follows: ethyl esters, alcohol acetates, fatty acids, C6 alcohols, whiskey lactones, vanillic derivatives, furanic derivatives, aldehydes, volatile phenols with positive and negative notes and sulfur compounds.

Soluble polysaccharides were determined and quantified (expressed in mg L$^{-1}$ of dextrans) using high-performance size exclusion chromatography coupled to a refractive index detector (HPSEC-RID) and following the methodology described in [8], with some modifications specified in [15]. Different polysaccharide fractions according to their molecular weight were obtained: high-molecular-weight (HMW) (range: 50–730 kDa), medium-molecular-weight (MMW) (range: 15–50 kDa), low-molecular-weight (LMW) (range: 9–15 kDa) and very low-molecular-weight (VLMW) (range: 5–9 kDa).

*2.4. Statistical Analyses*

All the variables analyzed were treated using the analysis of variance (ANOVA) and the least significant difference (LSD) test at the significance level of $p < 0.05$. Principal component analysis (PCA) was carried out with those variables that presented significant differences when using ANOVA. Statistical analyses were carried out using the Statgraphics Centurion XVIII statistical package.

## 3. Results

*3.1. Characterization of White Wines from Different PDOs*

3.1.1. Classic Enological Parameters, Glycerol and Organic Acids

Statistically significant differences by PDO were found in several of these parameters (Table 2). In the classic enological parameters, the clearest differences were observed in the ethanol content and total acidity value. Thus, the wines from RU (Rueda) (elaborated with the Verdejo and Sauvignon Blanc grape varieties) and CI (Cigales) (elaborated with the Verdejo grape variety) had a higher content of ethanol than the wines from the rest of the PDOs. However, the wines from RD (Ribera del Duero) (elaborated with Albillo) were characterized by higher total acidity than the other wines. This variable is highly associated with the content of organic acids [26], although this correlation was not observed in this study because the RD wines did not exhibit the highest values. The glycerol content was higher in the wines from RD and BI (Bierzo) than in those from RU. This parameter could have an impact on certain sensory properties of wines, such as body, viscosity and mouthfeel [33].

3.1.2. Phenolic Composition and Color Intensity

The RD wines exhibited the highest content of total tannins and total polyphenols (Table 2). According to the different groups of LMWPC analyzed, it was observed that HCATE, phenolic alcohols and HBA were the principal phenols (Table 2) in terms of concentration, similarly to other studies carried out on white wines [5,34]. The differences found between the wines depended on the phenolic group. The highest differences were found in the HCATE content, with the wines from RD having the highest content, mainly due to the differences found in trans-caftaric and trans-coutaric acids (Table S1). Lower differences were found in the HCA content, with the wines from RU having a higher content than those from TO due to their significantly higher content of both trans-caffeic and trans-*p*-coumaric acids (Table S1). On the contrary, the wines from TO exhibited a higher content of flavanols than those from CI, mainly due to the differences found in the content of catechin. Finally, the wines from RD and RU had a higher content of flavonols than those from BI due to the differences in quercetin glycosides. The phenolic content of wines is highly associated with the grape variety used, as well as with the environmental conditions of the vineyard, viticulture practices and the production techniques used (for example, oak wood and yeast lees aging) [1]. These compounds can influence the color intensity of wines due to their yellow color and/or oxidize them, increasing the color intensity values [34]. However, a clear correlation between the content of phenolic compounds and the color was not found because all the wines presented similar values of color intensity, with the exception of the TO wines that exhibited the lowest values.

**Table 2.** Classic enological parameters, color intensity, organic acids, glycerol, phenolic and volatile content of PDO white wines.

| | PDO | | | | | Grape Variety | | |
|---|---|---|---|---|---|---|---|---|
| | Ribera del Duero | Bierzo | Toro | Cigales | Rueda | Verdejo | Sauvignon blanc | Malvasía |
| **Basic parameters** | | | | | | | | |
| pH | 3.11 ± 0.11 [a] | 3.18 ± 0.10 [a] | 3.33 ± 0.16 [b] | 3.14 ± 0.06 [a] | 3.21 ± 0.13 [a] | 3.21 ± 0.14 [a,b] | 3.22 ± 0.09 [a,b] | 3.35 ± 0.22 [b] |
| Total acidity (g L$^{-1}$) | 5.99 ± 0.74 [b] | 5.37 ± 0.37 [a] | 5.16 ± 0.69 [a] | 5.64 ± 0.63 [a,b] | 5.29 ± 0.31 [a] | 5.24 ± 0.47 [a] | 5.46 ± 0.25 [a,b] | 5.38 ± 0.98 [a] |
| Ethanol (% vol.) | 12.4 ± 0.7 [a] | 12.1 ± 0.3 [a] | 11.9 ± 0.5 [a] | 13.1 ± 0.5 [b] | 13.8 ± 0.5 [c] | 13.3 ± 1.0 [b] | 13.5 ± 0.4 [b] | 11.8 ± 0.1 [a] |
| Organic acids (g L$^{-1}$) | 4.62 ± 0.63 [b,c] | 3.95 ± 0.45 [a] | 4.24 ± 0.65 [a,b] | 4.47 ± 0.08 [b,c] | 4.77 ± 0.38 [c] | 4.50 ± 0.32 [a] | 5.03 ± 0.42 [b] | 4.11 ± 0.75 [a] |
| Color intensity | 0.103 ± 0.026 [b] | 0.100 ± 0.000 [b] | 0.078 ± 0.017 [a] | 0.107 ± 0.024 [b] | 0.096 ± 0.017 [b] | 0.096 ± 0.019 | 0.092 ± 0.021 | 0.078 ± 0.024 |
| Glycerol (g L$^{-1}$) | 5.93 ± 0.54 [b] | 5.76 ± 0.56 [b] | 5.51 ± 0.50 [a,b] | 5.56 ± 0.98 [a,b] | 5.25 ± 0.38 [a] | 5.43 ± 0.57 [a,b] | 5.05 ± 0.19 [a] | 5.74 ± 0.64 [b] |
| **Polysaccharide composition (mg L$^{-1}$)** | | | | | | | | |
| High-molecular-weight (50 –730 kDa) | 110.8 ± 63.4 | 90.8 ± 45.3 | 122.7 ± 43.1 | 87.2 ± 22.8 | 91.9 ± 34.9 | 92.2 ± 31.8 | 108.1 ± 41.9 | 115.8 ± 56.5 |
| Medium-molecular-weight (15–50 kDa) | 77.3 ± 17.2 | 89.8 ± 24.4 | 81.2 ± 31.7 | 105.2 ± 38.7 | 94.3 ± 30.8 | 98.5 ± 50.0 | 71.6 ± 18.9 | 108.3 ± 28.8 |
| Low-molecular-weight (9–15 kDa) | 38.3 ± 19.3 [a,b,c] | 25.1 ± 10.7 [a] | 38.3 ± 15.0 [b,c] | 29.8 ± 11.5 [a,b] | 44.0 ± 12.6 [c] | 37.0 ± 12.5 | 45.4 ± 15.6 | 47.0 ± 13.7 |
| Very low-molecular-weight (5–9 kDa) | 21.3 ± 15.0 [b] | 11.9 ± 7.5 [a,b] | 20.2 ± 14.8 [b] | 20.3 ± 9.5 [b] | 7.2 ± 6.5 [a] | 11.6 ± 9.8 [a] | 8.7 ± 5.3 [a] | 27.8 ± 12.5 [b] |
| Total polysaccharides | 247 ± 86 | 218 ± 50 | 263 ± 66 | 243 ± 106 | 237 ± 84 | 239 ± 91 | 233.4 ± 49 | 299 ± 798 |
| **Phenolic composition (mg L$^{-1}$)** | | | | | | | | |
| Total polyphenols | 251 ± 78 [b] | 182 ± 31 [a] | 219 ± 61 [a] | 206 ± 58 [a] | 195 ± 23 [a] | 206 ± 39 [b] | 190 ± 19 [a] | 212 ± 88 [b] |
| Total tannins | 387 ± 174 [b] | 243 ± 62 [a] | 281 ± 101 [a] | 240 ± 55 [a] | 258 ± 62 [a] | 260 ± 54 | 237 ± 63 | 316 ± 151 |
| Hydroxybenzoic acids | 8.71 ± 7.45 | 7.42 ± 4.99 | 14.66 ± 8.65 | 9.56 ± 3.55 | 10.40 ± 4.17 | 13.52 ± 9.56 | 7.86 ± 4.54 | 6.51 ± 2.50 |
| Hydroxycinnamic acids | 2.63 ± 0.83 [a,b] | 2.83 ± 1.38 [a,b] | 2.58 ± 1.72 [a] | 2.81 ± 1.55 [a,b] | 4.88 ± 2.62 [b] | 4.27 ± 2.70 [b] | 4.32 ± 1.77 [b] | 1.36 ± 0.14 [a] |
| Hydroxycinnamic tartaric esters | 22.05 ± 8.10 [b] | 12.29 ± 7.21 [a] | 12.02 ± 8.24 [a] | 8.84 ± 4.55 [a] | 12.99 ± 5.01 [a] | 10.54 ± 4.90 | 14.40 ± 4.26 | 15.53 ± 11.64 |
| Flavanols | 3.75 ± 3.46 [a,b] | 4.17 ± 1.59 [a,b] | 6.11 ± 2.56 [b] | 3.61 ± 2.69 [a] | 5.28 ± 1.77 [a,b] | 5.36 ± 2.53 | 5.00 ± 1.18 | 4.83 ± 2.62 |
| Flavonols | 0.830 ± 0.432 [a,b] | 0.080 ± 0.136 [a] | 0.527 ± 0.645 [a,b] | 0.490 ± 0.575 [a,b] | 0.856 ± 0.490 [b] | 0.783 ± 0.884 | 0.493 ± 0.515 | 0.800 ± 0.817 |
| Phenolic alcohols | 19.0 ± 7.7 | 13.7 ± 5.4 | 13.2 ± 6.5 | 14.4 ± 6.8 | 12.4 ± 5.0 | 13.8 ± 5.7 [b] | 9.44 ± 1.64 [a] | 15.4 ± 8.2 [b] |
| **Volatile composition (µg L$^{-1}$)** | | | | | | | | |
| Higher alcohols | 243,535 ± 31,859 [a,b] | 247,147 ± 34,462 [a,b] | 286,215 ± 58,299 [b] | 254,661 ± 35,580 [a,b] | 232,003 ± 28,792 [a] | 246,706 ± 32,357 [a] | 228,076 ± 23,810 [a] | 312,262 ± 84,034 [b] |
| Ethyl esters | 2162 ± 445 [a] | 2130 ± 332 [a] | 2913 ± 845 [a,b] | 4167 ± 496 [b] | 3009 ± 564 [a,b] | 3237 ± 2317 | 2919 ± 365 | 3425 ± 233 |
| Alcohol acetates | 926 ± 771 [a] | 1126 ± 333 [a] | 3673 ± 2080 [b] | 1036 ± 714 [a] | 4176 ± 1445 [b] | 3231 ± 2003 | 4194 ± 1533 | 3726 ± 1996 |
| Σ ethyl esters and acetates | 3088 ± 542 [a] | 3256 ± 586 [a] | 6587 ± 2636 [b,c] | 5204 ± 4961 [b] | 7184 ± 1612 [c] | 6467 ± 3115 | 7113 ± 1571 | 7151 ± 2025 |
| C6 alcohols | 661 ± 45.3 [a] | 1384 ± 283 [b] | 1345 ± 479 [b] | 1592 ± 468 [b,c] | 1874 ± 676 [c] | 1626 ± 582 | 2015 ± 664 | 1370 ± 701 |
| Terpenes | 145 ± 39.2 [c] | 74.3 ± 21.6 [b] | 41.3 ± 32.2 [a] | 47.0 ± 8.0 [a] | 40.4 ± 14.4 [a] | 38.8 ± 21.4 | 47.7 ± 12.3 | 45.4 ± 19.4 |
| Whiskey lactones | 105 ± 79.4 [b] | 46.1 ± 24.0 [a,b] | 17.5 ± 9.1 [a] | 57.0 ± 28.8 [a,b] | 32.2 ± 22.7 [a,b] | 32.3 ± 18.8 | 47.1 ± 42.5 | 2.7 ± 1.6 |
| Vanillic derivatives | 96.9 ± 21.3 [a,b] | 89.1 ± 85.2 [a] | 135.9 ± 36.0 [b] | 121.7 ± 17.7 [a,b] | 96.4 ± 25.6 [a] | 107.7 ± 29.4 | 107.5 ± 32.1 | 133.3 ± 45.3 |
| Furanic derivatives | 1076 ± 148 [a] | 1113 ± 484 [a] | 1886 ± 908 [b] | 1223 ± 632 [a,b] | 1398 ± 578 [a,b] | 1500 ± 779 | 1321 ± 629 | 1824 ± 157 |
| Positive volatile phenols | 76.3 ± 65.1 [b] | 57.9 ± 47 [a,b] | 52.9 ± 47.6 [a,b] | 24.3 ± 17.9 [a] | 42.8 ± 25.5 [a,b] | 47.3 ± 36.5 | 28.7 ± 20.4 | 43.3 ± 16.1 |
| Fatty acids | 12,650 ± 891 [a,b] | 11,237 ± 1017 [a] | 13,161 ± 1902 [b] | 14,116 ± 3130 [b,c] | 14,805 ± 1036 [c] | 14,267 ± 1996 | 14,733 ± 1405 | 13,328 ± 1604 |
| Aldehydes | 21.6 ± 3.9 [a,b] | 20.6 ± 6.4 [a] | 34.4 ± 12.3 [c] | 32.4 ± 8.4 [b,c] | 25.2 ± 8.7 [a,b] | 29.6 ± 5.4 | 25.3 ± 6.4 | 31.6 ± 4.9 |
| Negative volatile phenols | 358 ± 73 [a] | 402 ± 237 [a] | 1079 ± 738 [b] | 320 ± 193 [a] | 277 ± 149 [a] | 464.8 ± 394 [a] | 196.1 ± 87 [a] | 1248 ± 842 [b] |
| Sulfur compounds | 17.8 ± 4.9 [a] | 16.7 ± 5.2 [a] | 32.4 ± 11.0 [b] | 14.8 ± 2.7 [a] | 14.5 ± 4.5 [a] | 17.8 ± 10.1 [a] | 15.6 ± 3.9 [a] | 33.9 ± 7.3 [b] |

Values with a different letter in the same row indicate statistically significant differences ($p < 0.05$), whereas values without a letter indicate no statistically significant differences.

### 3.1.3. Volatile Composition

Quantitatively, higher alcohols were largely the most important group, followed by fatty acids, ethyl esters and alcohol acetates and this result is in accordance with other studies carried out on other grape varieties [4,35]. In the case of higher alcohols, the most important differences were found between the wines from TO and RU, being higher in TO than in RU, due to the differences found in the content of isoamyl alcohol 3-methyl-1-butanol which was the main compound (Table S2). This fact was due to the highest content observed in the wines elaborated with the Malvasía grape variety coming from TO. Higher alcohols have largely been studied in wines due to their potential impact on the sensory profile. However, the literature about their positive or negative sensory effect is not unanimous because in some cases these compounds can increase the fruity and flowery notes and aromatic complexity, and in other cases they can mask the fruity perception and supply negative notes of fusel oil, solvent or malt [4,35,36] This literature reported that a total higher alcohol concentration below 300 mg $L^{-1}$ could contribute positively to the aromatic complexity of wines, and higher concentrations generate unpleasant aromas such as alcoholic, chemical and fusel notes [4]. The total content found in the studied wines was below this value, and they could supply positive sensory effects to the wines. Therefore, the TO wines showed higher aromatic complexity than RU wines. Statistically significant differences were also found in the concentrations of ethyl esters and alcohol acetates, which are related to the fruity aroma [37,38]. This group of volatile compounds was higher in the wines from RU, TO and CI than in those from RD and BI, mainly due to the differences found in the concentration of octanoate and decanoate ethyl esters and isoamyl acetate (Table S2), which were the main compounds. These compounds had a higher content than the odor threshold perception (2, 200 and 30 µg $L^{-1}$, respectively [37,38]) and for that reason the wines from RU, TO and CI could be characterized by having high notes of fruity aromas. On the contrary, the wines from RD and BI were characterized by a higher content of flowery aroma compounds (terpenes) than those from RU, TO and CI. The most important difference was found in the content of linalool (Table S2), with the RD and BI wines having values above the odor threshold perception (25 µg $L^{-1}$) [38]. The wines from RD had a higher content of C6 alcohols than the rest of the wines, mainly due to the differences in the content of 1-hexanol (Table S2), which is considered responsible for cut grass, herbaceous and resinous aromas [38]. However, the content of 1-hexanol seems not to have a sensory influence because the content found is below the odor threshold (8000 µg $L^{-1}$) [38]. With regard to fatty acids, the wines from RU, TO and CI presented a higher content with respect to those from BI (mainly due to the differences found in hexanoic and decanoic acids) (Table S2). These compounds could supply negative cheese, rancid, butter notes, etc. if the total concentration is above 10 mg $L^{-1}$ [37,38].

The principal differences in the aldehyde content were found in isobutyraldehyde and 3-methylbutanal (Table S2), which are considered to be oxidation markers [39]. The concentration of these compounds was above the odor threshold perception (6 and 4.6 µg $L^{-1}$, respectively), so they could supply negative sensory notes such as dried fruit and sweet fusel.

Some significant differences were found in the volatile compounds that come from oak wood. The wines from TO exhibited the highest content of vanillic and furanic derivatives while the wines from RD exhibited the highest content of whiskey lactones and positive volatile phenols. These differences could be mainly associated with the different types of oak wood barrels used for the fermentation of some of these wines. The concentration of vanillic and furanic derivatives did not exceed the odor threshold perception. On the other hand, the RD wines presented the trans-whiskey lactone content above the odor threshold perception (32 µg $L^{-1}$), which could have a positive impact on the sensory profile of these wines. In the case of positive volatile phenols, only guaiacol, eugenol and trans-isoeugenol had a content above the odor threshold (9.5, 6 and 6 µg $L^{-1}$, respectively). Therefore, they could have a sensory impact mainly in the RD and BI wines.

The TO wines had the highest content of some negative compounds, such as volatile phenols and sulfur compounds, and particularly those wines elaborated with the Malvasía grape variety. The differences were mainly due to the content of 4-vinylphenol and 4-vinylguaiacol, the compounds that contribute to the phenolic or medicinal flavor. Sulfur compounds, such as methyl thioacetate, ethyl thioacetate, dimethyl disulfide and methional exhibited the highest differences. These compounds supply a reduced aroma related to cooked/rotten vegetables and rotten eggs [40] and are formed through biological and chemical processes during the winemaking and storage processes, such as high turbidity of the must, a deficient nitrogen source, high temperature, and high addition of $SO_2$ to the grape must [41]. On the other hand, different conservation factors such as the contact time of the wine with lees and the storage time can also result in a higher presence of these compounds in wines [40].

### 3.1.4. Polysaccharide Composition

HMW polysaccharides were the most common (ranged between 35.8% and 46.7%), followed by the MMW (ranged between 31.2% and 43.2%), LMW (ranged between 11.5% and 18.6%) and VLMW ones (ranged between 3.0% and 8.6%) (Table 2). These percentages were very similar to those found in [6] in Chardonnay white wines. Only statistically significant differences were found in LMW and VLMW polysaccharides. These fractions are mainly composed of the polysaccharides from grape cell walls (RG-II) and, in lower proportion, of short chains of arabinogalactan proteins (AGP) and mannoproteins which come from grapes and yeast cell walls, respectively [6–8]. In this sense, the wines from RU had a higher content of LMW polysaccharides than those from BI. On the contrary, the wines from RU had a lower content of VLMW polysaccharides than those from RD, TO and CI. The higher content of these types of polysaccharides could increase the body and the mouthfeel complexity of wines and reduce the astringency, bitterness and hotness partially associated with the excessive content of ethanol [5,42,43].

### 3.2. Characterization of Rosé Wines from Different PDOs

### 3.2.1. Classic Enological Parameters, Glycerol and Organic Acids

Only significant differences were found in the ethanol content (Table 3), with the wines from RD having the highest values. As mentioned previously, differences in this parameter could have an effect on the sensory properties of wines [10–12] and contribute to their differentiation by PDO.

### 3.2.2. Phenolic Composition and Color Intensity

No significant differences were found in the content of total polyphenols, tannins and anthocyanins. The principal LMWPC were phenolic alcohols, followed by HCATE and HBA (Table 3). In general, the wines from RD had the highest content of phenolic alcohols (tyrosol and tryptophol) and HCA (trans-caffeic and trans-coumaric acids), although there were no statistically significant differences in all of the cases. On the other hand, the wines from TO exhibited a higher content of HCATE than those from BI (mainly due to the differences in trans-caftaric acid,). Finally, the wines from CI were characterized by the highest content of flavonols (quercetin) and their glycosides. LMWPC, especially flavonols, HCA and HCATE have been postulated as good copigments of anthocyanins of red and rosé wines and could enhance their red color due to an increase in their color intensity [44,45]. However, no significant differences were found in the color intensity of the wines studied.

**Table 3.** Classic enological parameters, color intensity, organic acids, glycerol, phenolic and volatile content of rosé wines from different PDOs.

| | PDO | | | |
|---|---|---|---|---|
| | **Ribera Del Duero** | **Bierzo** | **Toro** | **Cigales** |
| **Basic parameters** | | | | |
| pH | 3.37 ± 0.03 | 3.39 ± 0.18 | 3.30 ± 0.14 | 3.41 ± 0.09 |
| Total acidity (g L$^{-1}$) | 4.84 ± 0.37 | 4.81 ± 0.68 | 4.94 ± 1.05 | 4.54 ± 0.62 |
| Ethanol (% vol.) | 16.1 ± 0.2 [b] | 13.7 ± 1.4 [a] | 13.7 ± 1.2 [a] | 13.2 ± 1.7 [a] |
| Organic acids (g L$^{-1}$) | 4.06 ± 0.39 | 3.69 ± 0.53 | 3.68 ± 0.40 | 4.20 ± 0.62 |
| Color intensity | 0.735 ± 0.055 | 0.833 ± 0.653 | 0.897 ± 0.140 | 0.969 ± 0.360 |
| Glycerol (g L$^{-1}$) | 6.21 ± 0.34 | 6.07 ± 0.56 | 5.61 ± 0.96 | 5.60 ± 1.04 |
| **Polysaccharide composition (mg L$^{-1}$)** | | | | |
| High-molecular-weight (50–730 kDa) | 96.3 ± 47.5 | 118 ± 64.1 | 80.8 ± 15.2 | 80.2 ± 11.9 |
| Medium-molecular-weight (15–50 kDa) | 53.8 ± 23.4 [a] | 116.2 ± 31.9 [b] | 77.5 ± 57.8 [ab] | 82.6 ± 19.1 [ab] |
| Low-molecular-weight (9–15 kDa) | 14.8 ± 6.1 [a] | 48.2 ± 23.1 [b] | 25.8 ± 14.7 [a] | 48.5 ± 13.4 [b] |
| Very low-molecular-weight (5–9 kDa) | 10.3 ± 7.3 [ab] | 22.2 ± 20.2 [b] | 7.5 ± 9.6 [a] | 11.6 ± 8.7 [ab] |
| Total polysaccharides | 175 ± 29 [a] | 305 ± 108 [b] | 192 ± 71 [a] | 223 ± 30 [a] |
| **Phenolic composition (mg L$^{-1}$)** | | | | |
| Total polyphenols | 290 ± 48 | 326 ± 137 | 349 ± 56 | 370 ± 150 |
| Total tannins | 376 ± 91 | 449 ± 108 | 587 ± 82 | 513 ± 182 |
| Total anthocyanins | 25.0 ± 12.5 | 52.2 ± 46.8 | 48.5 ± 8.8 | 29.3 ± 17.1 |
| Hydroxybenzoic acids | 9.54 ± 7.59 [a] | 10.7 ± 7.1 [a] | 28.8 ± 12.4 [b] | 17.7 ± 8.3 [ab] |
| Hydroxycinnamic acids | 11.1 ± 5.19 [b] | 3.85 ± 2.16 [a] | 3.04 ± 0.83 [a] | 3.05 ± 1.82 [a] |
| Hydroxycinnamic tartaric esters | 21.4 ± 6.5 [ab] | 14.1 ± 6.5 [a] | 33.6 ± 14.1 [b] | 23.0 ± 10.7 [ab] |
| Flavanols | 7.95 ± 2.74 | 12.02 ± 8.76 | 7.53 ± 1.62 | 9.39 ± 2.99 |
| Flavonols | 1.88 ± 0.56 [a] | 1.00 ± 1.17 [a] | 1.62 ± 0.48 [a] | 3.72 ± 1.33 [b] |
| Phenolic alcohols | 42.3 ± 15.6 [c] | 18.7 ± 8.0 [a] | 38.7 ± 8.7 [bc] | 26.9 ± 11.9 [ab] |
| **Volatile composition (µg L$^{-1}$)** | | | | |
| Higher alcohols | 289,206 ± 32,407 [ab] | 280,482 ± 27,838 [ab] | 322,054 ± 35,152 [b] | 275,116 ± 56,012 [a] |
| Ethyl esters | 2425 ± 569 | 2470 ± 506 | 2307 ± 627 | 2426 ± 801 |
| Alcohol acetates | 4377 ± 1125 [b] | 2960 ± 1973 [a] | 2455 ± 1111 [a] | 2805 ± 1129 [a] |
| Σ ethyl esters and acetates | 6802 ± 1171 [b] | 5430 ± 2230 [ab] | 4880 ± 621 [a] | 5112 ± 1747 [ab] |
| C6 alcohols | 1345 ± 307 [a] | 1120 ± 242 [a] | 2193 ± 311 [b] | 1920 ± 365 [b] |
| Terpenes | 48.0 ± 9.5 [a] | 43.7 ± 18.7 [a] | 130 ± 58.0 [a] | 437 ± 198 [b] |
| Whiskey lactones | 12.3 ± 7.1 [a] | 12.0 ± 8.9 [a] | 55.2 ± 35.1 [a] | 605.2 ± 328.9 [b] |
| Vanillic derivatives | 49 ± 13.6 [a] | 70.2 ± 32.3 [a] | 131.5 ± 102.6 [a] | 676.3 ± 400.4 [b] |
| Furanic derivatives | 1222 ± 587 | 772 ± 245 | 1583 ± 827 | 1072 ± 633 |
| Positive volatile phenols | 28.8 ± 16.5 [a] | 36.2 ± 36.8 [a] | 19.8 ± 10.2 [a] | 316 ± 121 [b] |
| Fatty acids | 11,797 ± 377 [c] | 11,338 ± 1117 [bc] | 10,512 ± 1787 [b] | 9657 ± 1205 [a] |
| Aldehydes | 20.9 ± 4.8 | 24.5 ± 5.6 | 23.8 ± 4.7 | 27.0 ± 7.6 |
| Negative volatile phenols | 45.7 ± 32.5 [a] | 83.7 ± 57.9 [ab] | 20.0 ± 11.4 [a] | 183.4 ± 96.5 [b] |
| Sulfur compounds | 17.0 ± 5.3 | 15.5 ± 3.3 | 19.6 ± 7.7 | 18.3 ± 4.9 |

Values with a different letter in the same row indicate statistically significant differences ($p < 0.05$), whereas values without a letter indicate no statistically significant differences.

The individual content of LMWPC in the wines studied is indicated in Table S3.

### 3.2.3. Volatile Composition

Higher alcohols were the volatile group with the highest concentration in rosé wines, followed by fatty acids, ethyl esters and alcohol acetates. The higher alcohols content was significantly higher in the wines from TO than in those from CI, mainly due to the high content of 3-methyl-1-butanol. The total content in the wines from TO was higher than 300 µg L$^{-1}$ and, as mentioned before, could mask positive fruity aromas contributing to negative notes [4,35,36]

The RD wines were characterized by a higher content of alcohol acetates (isoamyl and β-phenylethyl acetates) than the rest of the wines, which could supply an improvement in the fruity and floral aroma profile because in both cases the content of these compounds

was above the odor threshold perception. Isoamyl acetate (30 µg L$^{-1}$) [37,38] could supply a fruity banana aroma and β-phenylethyl acetate (250 µg L$^{-1}$) [38] can contribute to rose floral notes. The CI wines presented the highest content of terpenic compounds due to the highest content of linalool. It was observed that all the wines studied exhibited the content of linalool above the odor threshold perception (25 µg L$^{-1}$) [38], but this content was significantly higher in the CI wines than in the rest and they could be characterized by having high floral notes. [38]. The TO and CI wines exhibited the highest concentrations of C6 alcohols that were mainly due to the content of 1-hexanol, a compound that is associated with the cut grass and herbaceous aromas [38]. However, as with white wines, this compound did not have an influence on the sensory profile of wines (content lower than its odor threshold perception). On the other hand, the wines from RD, BI and TO had a significantly higher content of fatty acids than those from CI, slightly higher than their threshold values (10 mg/L) [37,38].

Regarding the compounds that come from oak barrels which supply positive notes, it was observed that the wines from CI had the highest content of whiskey lactones (both cis- and trans-lactones), vanillic derivatives (vanillin, ethyl and methyl vanillates and acetovanillone) and positive volatile phenols (eugenol, trans-isoeugenol, syringol and 4-allylsyringol). In general, these compounds are responsible for supply wood, coconut, vanilla, spicy and smoky notes, improving the volatile complexity of wines [46,47]. These compounds could contribute to improving the sensory profile of the CI wines since their content was above the odor threshold perception (74 µg L$^{-1}$ for the cis-whiskey lactone [47], 32 µg L$^{-1}$ for the trans-whiskey lactone [47], 60 µg L$^{-1}$ for vanillin and 6 µg L$^{-1}$ for eugenol and trans-isoeugenol [48]). However, the CI wines also had a high content of negative volatile phenols, mainly, of 4-vinylguaiacol, which could cause undesirable aromas, such as smoky and curry notes, because their content was above the odor threshold perception (40 µg L$^{-1}$) [49]. These results indicate that the use of oak wood for aging rosé wines is more common in CI than in the rest of PDOs.

The individual concentration of the volatile compounds studied is reflected in Table S4.

### 3.2.4. Polysaccharide Composition

The content of total polysaccharides was the highest in the wines from BI. Significant differences were found in the content of LMW and VLMW polysaccharides, but also in the MMW ones, which are mainly composed of mannoproteins from yeast used in fermentation and AGPs from grapes [10]. The LMW and VLMW polysaccharide fractions are mainly composed of RG-II and short chains of mannoproteins and AGPs. As mentioned previously, a higher content of polysaccharides from grapes and yeast could have an important effect on the technological and sensory properties of wines [5–7,42]. These results were in accordance with those obtained in another study carried out on red wines from different PDOs [15] which also observed that the content of LMW polysaccharides in the wines from BI (elaborated with the Mencía grape variety) was significantly higher than in the wines from other PDOs (elaborated with the Tempranillo grape variety).

### 3.3. Multivariate Statistical Analyses

Principal component analysis (PCA) was carried out with the variables that exhibited statistically significant differences in the ANOVA with the objective of identifying the variables that contribute the most to the differentiation of wines according to their PDO. PCA results showed a very low percentage of the total variance explanation meaning low or no correlations between PCs, mainly in the case of white wines. Figure 1 shows the plane of the first two principal components of the scores of the white wines, which explained 31% of the total variance. These low percentages could be due to the fact that, although there is a differentiation in terms of the geographical area and climatic conditions, these PDOs are very close geographically. Furthermore, as commented previously, many of the wines studied in several PDOs were elaborated with the same grape variety (RU, CI and TO). Nevertheless, PC1 (17% of the total variance explained) allowed relative differentiation

of two different groups of wines: one formed by those from RU and CI (on the right side of the plane) and the other formed by those from RD, BI and TO (on the left side of the plane). However, the best differentiation was observed between the wines from RU and those from RD and BI. According to the loading values (Table 4), the variables that were most associated with differentiation of the wines from RU and CI were ethanol content, organic acids and several groups of volatile compounds such as alcohol acetates, fatty acids and C6 alcohols. On the other hand, the variables that allowed differentiation of the wines from RD, BI and TO were the glycerol and terpene content (more associated with RD and BI wines) and sulfur compounds (more associated with TO wines) (Table 4).

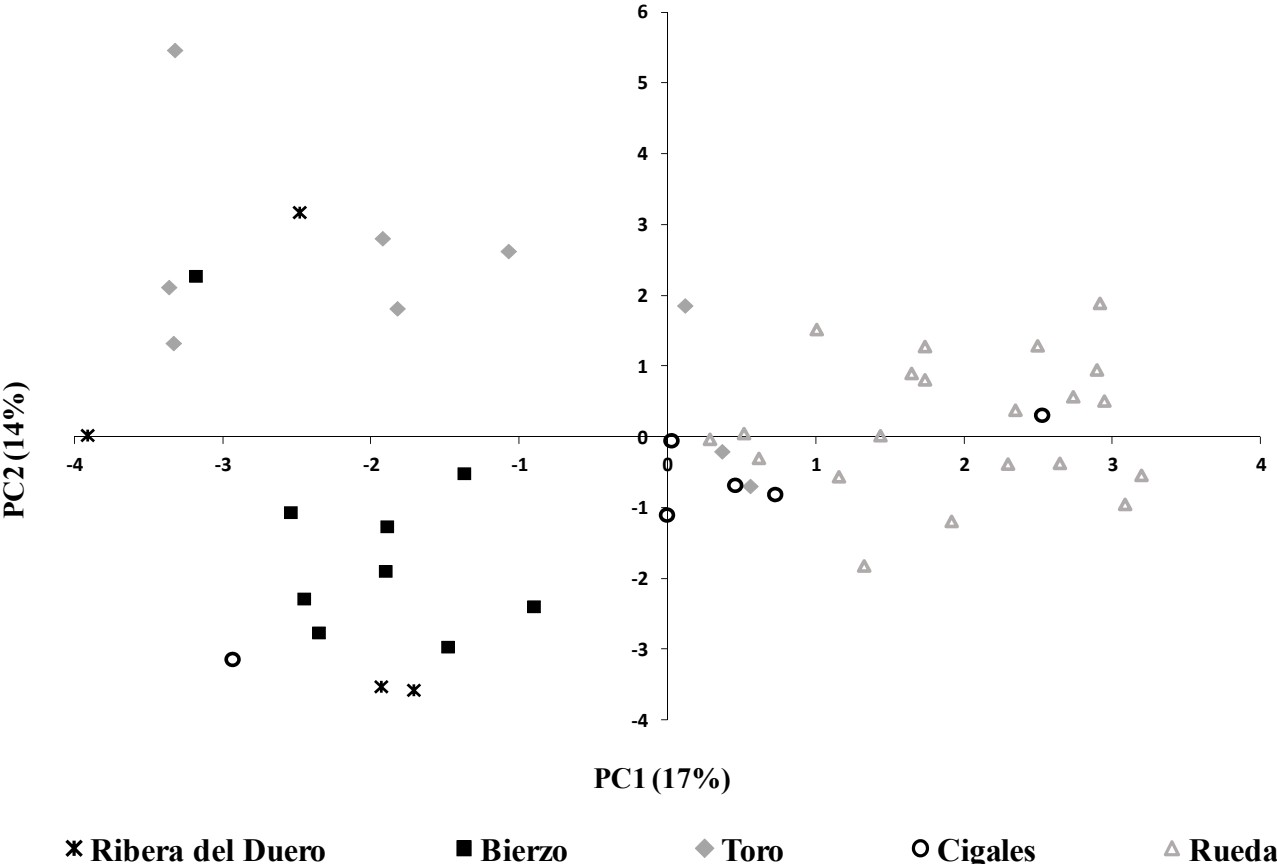

**Figure 1.** Distribution of white wines from different PDOs in the plot defined by the first two principal components (PC).

In the case of rosé wines, the first two PCs represented 45% of the total variance (Figure 2). PC1 explained 28% of the total variance and allowed differentiating between the wines from CI (on the right side of the plane) and the rest of the wines (on the left side of the plane). Groups of volatile compounds such as terpenes, vanillic derivatives, positive phenols and whiskey lactones had an important role in the differentiation of the wines from CI (Table 5). On the other hand, ethanol content and fatty acids were more associated with the differentiation of the wines from RD, BI and TO (Table 5). PC2 explained 17% of the total variance which mainly allowed clear differentiation between the wines from BI and RD, with LMW and VLMW polysaccharides highly associated with the wines from BI (Table 5).

**Table 4.** PCA loading values of the variables selected in white wines.

| Variables | PC1 | PC2 |
|---|---|---|
| **pH** | | **0.523** |
| **Total acidity** | | −0.463 |
| **Ethanol** | **0.714** | |
| **Organic acids** | **0.576** | |
| **Color intensity** | | |
| **Glycerol** | **−0.595** | −0.362 |
| **LMW polysaccharides** | 0.259 | 0.456 |
| **VLMW polysaccharides** | −0.467 | |
| **Total polyphenols** | −0.314 | **0.557** |
| **Total tannins** | −0.287 | 0.380 |
| **Hydroxycinnamic acids** | 0.399 | |
| **Hydroxycinnamic tartaric esters** | | 0.424 |
| **Flavanols** | | **0.563** |
| **Flavonols** | | **0.502** |
| **Higher alcohols** | −0.443 | 0.485 |
| **Ethyl esters** | 0.336 | |
| **Alcohol acetates** | **0.586** | **0.544** |
| **C6 alcohols** | **0.529** | |
| **Terpenes** | **−0.528** | −0.317 |
| **Whiskey lactones** | −0.392 | |
| **Vanillic derivatives** | −0.250 | **0.542** |
| **Furanic derivatives** | | 0.321 |
| **Positive volatile phenols** | −0.388 | |
| **Fatty acids** | **0.646** | |
| **Aldehydes** | | 0.242 |
| **Negative volatile phenols** | −0.421 | **0.617** |
| **Sulfur compounds** | **−0.633** | |

Bold numbers indicate the loading values that contributed the most to each principal component (PC). Loadings with the absolute value below 0.250 are not shown. LMW: low molecular weight; VLMW: very low molecular weight.

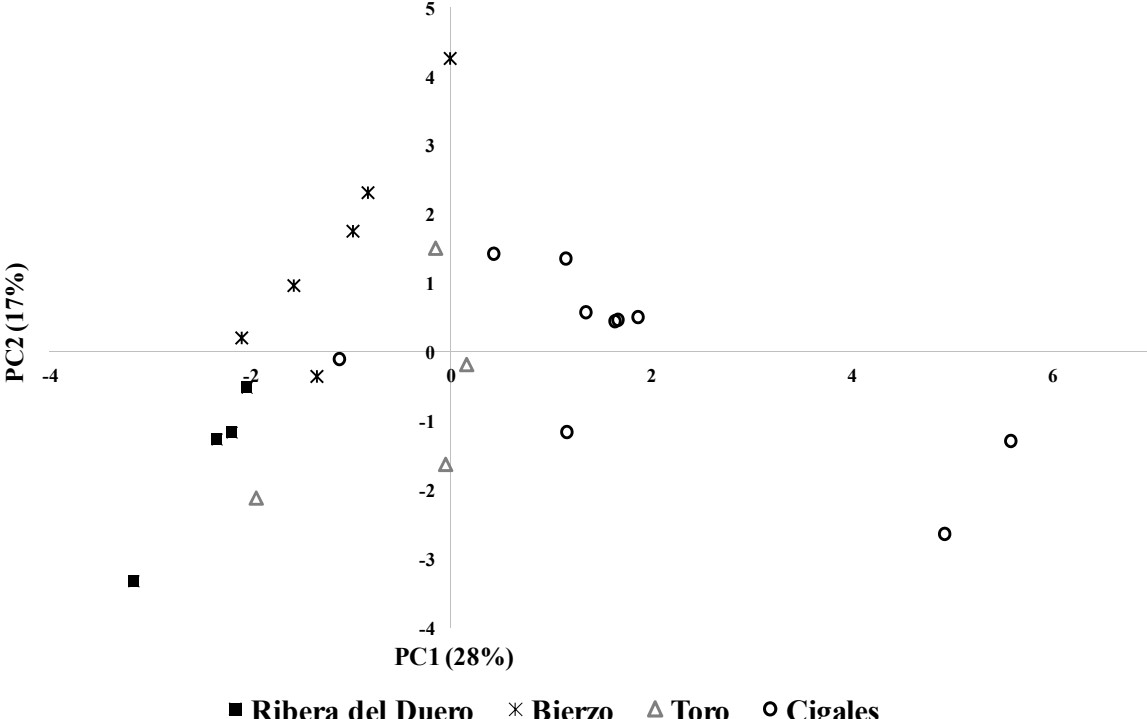

**Figure 2.** Distribution of rosé wines from different PDOs in the plot defined by the first two principal components (PC).

**Table 5.** PCA loading values of the variables selected in rosé wines.

| Variables | PC1 | PC2 |
|---|---|---|
| Ethanol | **−0.606** | −0.423 |
| MMW polysaccharides | | **0.890** |
| LMW polysaccharides | 0.486 | **0.767** |
| VLMW polysaccharides | | **0.633** |
| Hydroxybenzoic acids | | −0.344 |
| Hydroxycinnamic acids | −0.328 | −0.416 |
| Hydroxycinnamic tartaric esters | | |
| Flavonols | 0.463 | |
| Phenolic alcohols | | −0.488 |
| Alcohol acetates | −0.370 | |
| C6 alcohols | 0.448 | |
| Terpenes | **0.913** | |
| Whiskey lactones | **0.689** | −0.376 |
| Vanillic derivatives | **0.871** | |
| Positive volatile phenols | **0.817** | −0.347 |
| Fatty acids | **−0.652** | |
| Negative volatile phenols | 0.433 | |

Bold numbers indicate the loading values that contributed the most to each principal component (PC). Loadings with the absolute value below 0.250 are not shown. MMW: medium molecular weight; LMW: low molecular weight; VLMW: very low molecular weight.

## 4. Conclusions

Several volatile and nonvolatile variables contributed, to a greater or lesser extent, to the differentiation of the studied white and rosé wines from the different PDOs located very close geographically. The white wines from RU and CI were characterized by the highest content of ethanol, while the wines from RD and BI by the highest content of glycerol, compounds that can affect gustatory attributes. The wines from RD and BI were characterized by a high terpenic content providing floral notes to these wines, while the wines from RU, TO and CI were characterized by a high prevalence of fruity aromas supplied by ethyl esters and alcohol acetates.

Clear differences were also found between the rosé wines, with the wines from RD being the most alcoholic ones. The wines elaborated with the Mencía grape variety from BI were characterized by the highest polysaccharidic content, which could have a positive sensory effect on the mouthfeel. The wines from CI were characterized by their volatile profile complexity, having the highest content of volatile compounds from the oak wood, terpenes and C6 alcohols which provide pleasant woody, floral and herbaceous aromas. On the contrary, the RD wines were the richest in alcohol acetates responsible for fruity aromas.

According to the obtained results, other factors such as winemaking techniques used in the region and/or in the winery could have an influence on wine composition. Similar studies should be carried out including a larger number of sample wines, considering other variables, such as price and category, and evaluating sensory attributes to establish the relationship between compounds and sensory characteristics of the wines.

**Supplementary Materials:** The following are available online at https://www.mdpi.com/article/10 .3390/beverages7030049/s1, Table S1: Concentration of individual low molecular weight phenolic compounds (mg/L) identified and quantified in the white wines. Table S2: Concentration of individual volatile compounds (μg/L) identified and quantified in the white wines. Concentration of individual low molecular weight phenolic compounds (mg/L) and volatile compounds (μg/L) identified and quantified in the rosé wines. Concentration of individual volatile compounds (μg/L) identified and quantified in the rosé wines. Table S3: Concentration of individual low molecular weight phenolic compounds (mg/L) and volatile compounds (μg/L) identified and quantified in the rosé wines. Table S4: Concentration of individual volatile compounds (μg/L) identified and quantified in the rosé wines.

**Author Contributions:** Conceptualization, S.P.-M., P.L.-d.-l.-C.; formal analysis, R.D.B.-G., M.B.-H., H.d.V.-H.; statistical analysis, R.D.B.-G.; supervision, S.P.-M., P.L.-d.-l.-C.; writing—original draft preparation, R.D.B.-G., S.P.-M.; writing—review and editing, S.P.-M., R.D.B.-G.; project administration, S.P.-M.; funding acquisition, S.P.-M. All authors have read and agreed to the published version of the manuscript.

**Funding:** This study was supported by project 2017/721 of the Rural Development Program (PDR) of Castile and León 2014–2020 and financed with the FEADER funds.

**Acknowledgments:** The authors are grateful to the different regulatory councils for providing the wines for the study.

**Conflicts of Interest:** The authors declare no conflict of interest.

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
