# Peer review of "Volatile and Non-Volatile Characterization of White and Rosé Wines from Different Spanish Protected Designations of Origin"

_beverages, doi:10.3390/beverages7030049_

Round 1
Reviewer 1 Report
General comments
Through an experimental study, the research paper aims to characterize white and rosè wines produced within several Spanish Protected Denominations of Origin (PDOs), paying attention to analyzing the volatile and non-volatile compounds in order to differentiate.
The major strength of this study is that differentiation among PDOs is a relevant topic in wine literature and business. In this perspective, the paper assessed key analysis to characterize their profile complexity. However, the introductory part of the paper should be better framed in the competitive scenario concerning PDOs’ wine market while some sentences showed some oversights, which have been thoroughly listed below.
Relationship to Literature
In order to introduce the topic of the paper, it might be useful to include, in the introductory section, some references about the relationship between the concept of PDOs’ differentiation for valuing the Geographical Indications and the concept of differential qualities of wines as a crucial issue for sustaining their market:
- Bernabéu, R., Brugarolas, M., Martínez‐Carrasco, L., & Díaz, M. (2008). Wine origin and organic elaboration, differentiating strategies in traditional producing countries. British Food Journal.
- Galletto, L., Caracciolo, F., Boatto, V., Barisan, L., Franceschi, D., & Lillo, M. (2021). Do consumers really recognise a distinct quality hierarchy amongst PDO sparkling wines? The answer from experimental auctions. British Food Journal, 123(4), 1478-1493. doi:10.1108/BFJ-07-2020-0625.
- Dressler, M. (2018). The German Wine Market: A Comprehensive Strategic and Economic Analysis. Beverages, 4(4), 92. Retrieved from https://www.mdpi.com/2306-5710/4/4/92.
Conclusions
The limitations of the study should be addressed. Moreover, the mention of a scope for further research on the same topic seems to be missing.
Quality of communication
The quality of the presentation is good. Even though the English language style is very readable, it needs a last revision.
Along with the paper, there are some oversights and uses of terminology that are not adequate and should require final refinement of the paper and a careful check of references:
- Pag. 1, from lines 16 onwards. The abstract does not describe the statistical methods used.
- Pag. 1, lines 17-19. Acronyms of different Spanish PDOs are introduced in the abstract without being previously defined (only then from pag. 3).
- Pag. 1, lines 26-27. It is recommended to use keywords other than those used in the title to increase the visibility of the paper.
- Pag. 6, paragraphs 3.2.2 and 3.2.3. Please check and delete for double parenthesis along with the text.
- Pag. 6, paragraphs 3.2.2 and 3.2.3. Please check for repetition of table reference along the text: e.g. (table S3)), (table S4)).
- Pag. 6, line 295. I would suggest changing the reference term 'our group' to the third person.
- Pag. 8, lines 327-329 and pag. 9, lines 332-334. It may be appropriate to introduce a general category defining the first six parameters in the tables. In addition, carefully check for consistency of both table formatting and number decimal separators.
Comments to the Author
I believe that the effort produced by the author/s in this interesting field of research can be recognised after a minor review.
Author Response
Dear reviewer,
Thank you very for your comments that will contribute to improve this manuscript. Here, we have responded your comments point by point:
Relationship to Literature
In order to introduce the topic of the paper, it might be useful to include, in the introductory section, some references about the relationship between the concept of PDOs’ differentiation for valuing the Geographical Indications and the concept of differential qualities of wines as a crucial issue for sustaining their market. We improve the introduction section (page 1) including the references suggested by the reviewer.
Conclusions
The limitations of the study should be addressed. Moreover, the mention of a scope for further research on the same topic seems to be missing. It has been included a paragraph that include the possible future studies that addressing the limitations of the present study and allowed to clarify the variables that differentiate wines from different PDOs.
Even though the English language style is very readable, it needs a last revision. The manuscript has been reviewed by an English speaking person. We have sent the manuscript to an English person review it.
Quality of communication
- Pag. 1, from lines 16 onwards. The abstract does not describe the statistical methods used. We have included in the abstract the statistical methods used.
- Pag. 1, lines 17-19. Acronyms of different Spanish PDOs are introduced in the abstract without being previously defined (only then from pag. 3). We have included them.
- Pag. 1, lines 26-27. It is recommended to use keywords other than those used in the title to increase the visibility of the paper. We have changed some of the keywords.
- Pag. 6, paragraphs 3.2.2 and 3.2.3. Please check and delete for double parenthesis along with the text. We have checked and deleted it.
- Pag. 6, paragraphs 3.2.2 and 3.2.3. Please check for repetition of table reference along the text: e.g. (table S3)), (table S4)). We have deleted them and have included a new sentences referring to these tables.
- Pag. 6, line 295. I would suggest changing the reference term 'our group' to the third person. We have changed it.
- Pag. 8, lines 327-329 and pag. 9, lines 332-334. It may be appropriate to introduce a general category defining the first six parameters in the tables. In addition, carefully check for consistency of both table formatting and number decimal separators. We proposed a general category named "basic parameters". In addition, we have checked the tables mentioned.
Thank you very much.

Reviewer 2 Report
The characterization of white and rosé wines was based on the usual volatile and non-volatile profiles and classical enological parameters. The paper is generally well written, and the results are well discussed. However, the weakness of this work is the absence of sensory analysis of wines, to establish the relationship between compounds and organoleptic characteristics of the wines.
Author Response
Dear reviewer,
Thank you very much for your comments.
Dear reviewer,
Thank you very much for your comments.
The characterization of white and rosé wines was based on the usual volatile and non-volatile profiles and classical enological parameters. The paper is generally well written, and the results are well discussed. However, the weakness of this work is the absence of sensory analysis of wines, to establish the relationship between compounds and organoleptic characteristics of the wines.
It is true that the great weakness of this study is the absence of sensory analysis. We have reflected it in the conclusions in this new revision, and we hope to be able to address it in future investigations.

Reviewer 3 Report
Dear Authors,
I had some difficulty in reviewing this manuscript due to many analytical variables as shown in Table 2, although this study aimed to characterize white and rose wines produced from geographically close areas with different grape varieties in Spain. The multivariate statistical analysis seems not to provide much helpful information to understand the characters of the wines being investigated since each of these input variables have different significance and weight. Therefore, it would be much better and clear to understand if the objectives are focused on important factors that can easily characterize the wines.
Below are some comments and suggestions:
- Introduction: Although the scope of the study is wide, the introduction is insufficient. In particular, there is no background introduction to the volatile aromas, which is very important for the characterization of wines.
- Analytical Methods (line 100-104): "Minor" is not a good expression about volatile aroma compounds as they are an important factor in characterizing wines. "For each sample.......", this method description is not clear, no chemical name and preparation of stock solution for the internal standard. Furthermore, there is no explanation for quantification. In general, the internal standard method describes the preparation of the stock solution followed by spiking into the sample solution and the final concentration in sample wine.
- Minor spell error (line 107): "inyector"---->injector
- Volatile composition (line 262-285): It is the most difficult part of the study to interpret the result from the analysis of volatiles. Therefore, please pay more careful attention to interpret the result, not just by PCA data, but consider the sensory thresholds of each volatile compound. PCA doesn't tell the significance of specific volatile compounds. They just separate variables and objects by physical numbers.

Author Response
Dear reviewer,
Thank you very much for your comments that will improve the quality of this manuscript. Here, we attached the reply to your comments:
- Introduction: Although the scope of the study is wide, the introduction is insufficient. In particular, there is no background introduction to the volatile aromas, which is very important for the characterization of wines. We have improved the introduction with a new paragraph focused, mainly, in the volatile aromas.
- Analytical Methods (line 100-104): "Minor" is not a good expression about volatile aroma compounds as they are an important factor in characterizing wines. "For each sample.......", this method description is not clear, no chemical name and preparation of stock solution for the internal standard. Furthermore, there is no explanation for quantification. In general, the internal standard method describes the preparation of the stock solution followed by spiking into the sample solution and the final concentration in sample wine.We deleted the expression "minor" and we have only left "volatile compounds". On the other hand, we have explained with more details the preparation of the solution of internal standards.
- Minor spell error (line 107): "inyector"---->injector. We have changed it and we have revised other spelling errors.
- Volatile composition (line 262-285): It is the most difficult part of the study to interpret the result from the analysis of volatiles. Therefore, please pay more careful attention to interpret the result, not just by PCA data, but consider the sensory thresholds of each volatile compound. PCA doesn't tell the significance of specific volatile compounds. They just separate variables and objects by physical numbers. We have improved this section focusing the comments in the odor threshold of the most important compounds.

Round 2
Reviewer 3 Report
Dear Authors,
The followings are some of review result of the manuscript. Please consider them to revise the manuscript.
Line 88-89: This descriptions about the gases used in GC-FID are not necessary. Please refer to other well written papers similar to this type of flavor studies.
Line 116-120: The purpose of using the internal standard is to quantify volatile components in samples. Therefore, the internal standard added to the sample must use components that aren't present in the sample. Please check whether the internal standards being used in your experiments are included in the samples.
Line 128-129: The range of mass (40-200) for scanning or SIM is too low. Please check mass of volatiles found in your samples.
Line 213-214, line 298-307, line 323-325: Please list the references for the threshold values that you mentioned. These values should reasonably be representing your analytical results as there is a tremendous differences in getting the values, depending on evaluation conditions.
PCA results: Although the sample groups are visually separated, all these PCA results show very low percentage of explanation among variances, meaning low or no correlations between PCs. Please explain these low percentages with regards to the separation.

Author Response
Dear reviewer,
Thank you again for your comments. Here, we attached the replies for these comments:
Line 88-89: This descriptions about the gases used in GC-FID are not necessary. Please refer to other well written papers similar to this type of flavor studies. We deleted this sentence according to other similar papers.
Line 116-120: The purpose of using the internal standard is to quantify volatile components in samples. Therefore, the internal standard added to the sample must use components that aren't present in the sample. Please check whether the internal standards being used in your experiments are included in the samples. The internal standards used were not present in the studied samples, and we have added a sentence in text.
Line 128-129: The range of mass (40-200) for scanning or SIM is too low. Please check mass of volatiles found in your samples. We have checked the mass of volatiles found in the samples and this range is sufficient for the volatile compounds analyzed.
Line 213-214, line 298-307, line 323-325: Please list the references for the threshold values that you mentioned. These values should reasonably be representing your analytical results as there is a tremendous differences in getting the values, depending on evaluation conditions. We have listed the references for the odor threshold values mentioned. We have included a new reference for justify the odor thresholds mentioned.
PCA results: Although the sample groups are visually separated, all these PCA results show very low percentage of explanation among variances, meaning low or no correlations between PCs. Please explain these low percentages with regards to the separation. We have explained in the text the possible aspects of these low percentages.
Thank you very much.
Kind regards.
Rubén del Barrio Galán.
